# Selective Excitation of Lanthanide Co-Dopants in Colloidal Lead-Free Halide Perovskite Nanocrystals as a Multilevel Anti-Counterfeiting Approach

**DOI:** 10.3390/nano15241838

**Published:** 2025-12-05

**Authors:** Olexiy Balitskii, Wilson Kagabo, Pavle V. Radovanovic

**Affiliations:** Department of Chemistry, University of Waterloo, 200 University Avenue W., Waterloo, ON N2L3G1, Canada; wilson.kagabo@uwaterloo.ca

**Keywords:** lead-free double perovskite, lanthanide phosphorescence, selective excitation

## Abstract

Doping lead-free halide perovskite nanocrystals with trivalent lanthanide ions has emerged as a promising strategy for engineering their optical properties in various photonic applications. Here, we report the design and synthesis of a series of lead-free double halide perovskite (Cs_2_Na(In/Y/Gd)Cl_6_) nanocrystals co-doped with a pair of different lanthanides (e.g., Tb^3+^, Dy^3+^, and Eu^3+^) as emission centers, and ns^2^ ions (Sb^3+^ or Bi^3+^) as sensitizers. The tunability of the delayed photoluminescence spectral density was achieved through the selective excitation of lanthanide dopants either via ligand-to-metal charge transfer (e.g., Eu^3+^) or via ns^2^ ion s-p transitions (e.g., Dy^3+^ or Tb^3+^). The intensities of the narrow lanthanide f-f emission bands can, therefore, be tuned by modulating the excitation wavelength and/or dopant ratio, allowing for the accurate engineering of the emission color coordinates and spectral density. We also demonstrated time-resolved tuning of the photoluminescence spectral density for the investigated nanocrystal host lattices co-doped with transition-metal (Mn^2+^) and lanthanide ions, owing to a large difference between the decay dynamics for Mn^2+^ d-d and lanthanide f-f transitions. The rational co-doping of double halide perovskite nanocrystals reported in this work provides a new strategy for generating pre-designed multilevel luminescent signatures for protection against counterfeiting.

## 1. Introduction

Trivalent lanthanide ions (Ln^3+^) have been recognized as efficient and versatile light emitters and fluorescent probes owing to their characteristic intra-ionic f-f transitions in the UV (e.g., cerium, gadolinium, praseodymium), visible (e.g., terbium, europium, dysprosium), and near-infrared (e.g., erbium, ytterbium, neodymium) regions. Their major drawbacks, however, are very small excitation cross sections and the effective quenching of photoluminescence (PL) by coordinated ligands [1,2]. These disadvantages have been overcome by embedding lanthanide ions into host matrices, including oxides [3,4], oxyhalides [5], zinc chalcogenides [6], and halide perovskites [7], providing efficient energy transfer to the lanthanide ions. Double halide perovskites (DHPs) having the formula unit A_2_B(I)B(III)X_6_ (A = Cs, Rb; B(I) = Ag, Na; B(III) = In, Sb, Bi, Sc, Y; X = Cl, Br, I) are particularly attractive as host lattices for lanthanides, since Ln^3+^ ions are isovalent with respect to the octahedral B(III) sites; furthermore, a wide range of cation substitutions and desirable cation combinations [8] can be attained in DHPs by balancing cubic cell stretching and shrinking. Hence, a considerable number of photonic applications [9,10] can be implemented by utilizing different Ln^3+^ ions in halide perovskites, such as X-ray luminescence for high-energy radiation imaging [11], electroluminescence for light-emitting diodes (LEDs) [12], and anti-Stokes luminescence for upconversion bioprobes and solar cell near-infrared (NIR) sensitization [13]. Using characteristic lanthanide and/or transition metal PL is particularly relevant for anticounterfeiting and encryption applications, as it offers multi-level signatures via the selective excitation of Ln^3+^ f-f, transition metal d-d, and self-trapped exciton (STE) emissions in both the steady-state and afterglow time regimes [14]. The luminescence of these materials consists of several spectral bands, which may not be desirable for monochromatic light-emitting applications (e.g., lasing), but it is advantageous for encoding purposes or white light generation. Most proof-of-concept studies involving complex DHP lattices have been performed for bulk (microcrystal) samples [14,15,16,17,18,19,20,21]. However, encoding unique patterns with high spatial resolution requires precise pixelation on a given substrate. In this context, the design and synthesis of stable colloidal multi-cation-doped DHP nanocrystals (NCs) could enhance the applicability of these materials through inkjet and roll-to-roll printing on thin and flexible substrates, or by embedding NCs into polymeric and plastic materials [22]. Developing new multi-lanthanide-based NC emitters also offers opportunities not available with analogous microcrystal emitters, including the potential to achieve photostable stimulated emission depletion nanoscopy labels [23]. Furthermore, NCs exhibit almost negligible optical density in the visible spectrum, which is beneficial for, e.g., NC-containing contact lenses that employ the selective excitation strategy of Ln^3+^ co-doping to perceive the invisible parts of the electromagnetic spectrum via remapping visible red-green-blue (RGB) Ln^3+^ emissions [24].

In this work, we demonstrate the ability to manipulate the PL spectral density and peak intensity ratios for different lanthanide and transition metal co-doped DHP NCs using excitation wavelength and time delay as tuning parameters. Owing to the different excitation pathways and excited-state lifetimes for lanthanide (europium, terbium, and dysprosium) and transition metal (manganese) dopants in Cs_2_NaInCl_6_ (CNIC), Cs_2_NaYCl_6_ (CNYC), and Cs_2_NaGdCl_6_ (CNGC) DHP NC host lattices, a precise tuning of the Ln^3+^ (f-f) and Mn^2+^ (d-d) emission intensities can be achieved in both frequency and time domains. A comparative investigation of these emission signatures within a series of DHP NCs enables a detailed understanding of the dopant excitation and energy-transfer mechanisms. The ability to tune spectral density throughout the visible range by adjusting different parameters in a pre-designed manner renders these complex NC systems promising for multi-level protection against counterfeiting.

## 2. Materials and Methods

### 2.1. Materials

Cesium carbonate (Cs_2_CO_3_, 99%), gadolinium acetate hydrate (Gd(OAc)_3_ × H_2_O, 99.9%), yttrium acetylacetonate hydrate (Y(acac)_3_ × H_2_O, 99.9%), erbium chloride anhydrous (ErCl_3_, 99.9%), terbium chloride hexahydrate (TbCl_3_ × 6H_2_O, 99.9%), europium nitrate hexahydrate (Eu(NO_3_)_3_×6H_2_O, 99.9%), and manganese acetylacetonate (Mn(acac)_2_, 95%) were purchased from Strem Chemicals (Newburyport, MA, USA). Indium acetate (In(OAc)_3_, 99.99%), silver nitrate (AgNO_3_, 99.9+%), bismuth chloride (BiCl_3_, 99.99%), antimony acetate (Sb(OAc)_3_, 97%), oleic acid (OA, 90%), and 1-octadecene (ODE, 90%) were received from Thermo FisherScientific (Waltham, MA, USA). Dysprosium nitrate hydrate (Dy(NO_3_)_3_ × H_2_O, 99.9+%), samarium nitrate hexahydrate (Sm(NO_3_)_3_×6H_2_O, 99.999%), hydrochloric acid (37%), oleylamine (OLA, 70%), and sodium acetate (NaOAc, 99.0%) were received from Sigma Aldrich (Darmstadt, Germany).

### 2.2. Nanocrystal Synthesis

All syntheses were carried out according to the modified procedure we have previously reported [25]. Specifically, cesium/sodium oleate was prepared though mixing of cesium carbonate (1.628 g, 5 mmol), sodium acetate (0.39 g, 4.75 mmol), ODE (20 mL), and OA (5 mL) in a 50 mL two-neck round bottom flask, degassing the mixture under vacuum at 115 °C for one hour and then heat-treating under argon at 150 °C until the solution became clear. The obtained oleate solidifies at room temperature and thus requires preheating before injection. B(III) cation precursors were combined in a total of 0.3 mmol (e.g., 0.21 mmol of indium acetate, 0.015 mmol of antimony acetate, 0.0375 mmol of europium nitrate, and 0.0375 mmol of dysprosium nitrate for 5% (Sb)/12.5% (Eu)/12.5% (Dy) CNIC sample; 0.21 mmol of yttrium acetylacetonate, 0.015 mmol of antimony acetate, 0.0375 mmol of europium nitrate, and 0.0375 mmol of dysprosium nitrate for 5%/12.5%/12.5% CNYC sample; 0.21 mmol of gadolinium acetate, 0.015 mmol of antimony acetate, 0.0375 mmol of europium nitrate, and 0.0375 mmol of dysprosium nitrate for 5%/12.5%/12.5% CNGC sample. When specifying compositions, “%” refers to molar percentages for the co-doped B(III) ions in the host lattice. B(III) precursors were then mixed with AgNO_3_ (0.01 mmol), ODE (15 mL), OA (1.5 mL), OLA (1.5 mL), and hydrochloric acid (0.45 mL), loaded into a 50 mL three-neck round-bottom flask, stirred, and degassed under vacuum at 115 °C for one hour. Subsequently, under nitrogen, the temperature was raised to 200 °C, and 1 mL of Cs/Na-oleate solution was rapidly injected. The reaction was allowed to run for 5 min and was then quenched in an ice/water bath. The obtained mixture was centrifuged for 5 min at 7800 rpm, and the supernatant was discarded. The residue was redispersed by adding 15 mL of toluene or hexane, then centrifuged for 1 min at 900 rpm. The supernatant was collected and filtered through a 0.2 μm PTFE filter.

### 2.3. Measurements and Characterization

Transmission electron microscopy (TEM) images and selected-area electron diffraction (SAED) patterns were recorded using a JEOL JEM-200 electron microscope (*JEOL* Ltd., Tokyo, Japan) operated at 200 kV. The NC specimens were drop-cast onto 400-mesh carbon-plated copper grids and dried. Before the TEM/SAED measurements, the organic residues from the samples were partially removed by argon plasma treatment using a Targeo EM Plasma Cleaner from PIE Scientific (Union City, CA, USA). A Rigaku MiniFlex II X-ray diffractometer (*Rigaku*, Tokyo, Japan) equipped with a Cu Kα X-ray source and a scintillation detector was employed for collecting XRD patterns.

The absorption spectra of colloidal DHP NCs were acquired with a Varian Cary 5000 UV−vis-NIR spectrophotometer (Varian, Agilent Technologies Deutschland GmbH, Waldbronn, Germany) using 10 mm path-length two-polished-window quartz cuvettes. Steady-state photoluminescence excitation (PLE) and emission (PL) spectra of DHP NCs were collected with a Varian Cary Eclipse spectrophotometer using 10 mm quartz four-polished-window cuvettes. The STE PL/PLE spectra were recorded in the fluorescence mode, while the Ln^3+^ and Mn^2+^ PL/PLE spectra were recorded in the phosphorescence mode with a 200 μs delay time and a 5 ms gate time, using a Xenon flash lamp as the excitation source. Time-resolved photoluminescence (TRPL) data for Ln^3+^/Mn^2+^ emissions were collected using the same instrument (Varian Cary Eclipse (Santa Clara, CA, USA)).

## 3. Results and Discussion

Colloidal DHP NCs were synthesized by the hot-injection method, as described in the Materials and Methods Section. To enhance distortions in the native DHP lattice, which enables free exciton localization and breaking of the parity forbidden nature of the band edge transition [26], we added a small amount of silver to the B(I) site. The presence of an ns^2^ ion (antimony or bismuth) in the B(III) site of the host lattice ensures an alternative excitation pathway based on intra-ionic s-p transition for STE and lanthanide ions. Figure 1 and Appendix A provide an overall assessment of the morphology, size, and structure of the NCs. Overview TEM images of Ln^3+^-doped DHP NCs (Appendix A) indicate that the CNYC and CNIC NCs generally have a bar-like morphology. In contrast, CNGC NCs have more irregular shapes with rounded edges. Furthermore, high-resolution TEM images of individual NCs in Figure 1a–c show well-resolved lattice fringes, confirming their single-crystalline nature. For cases where the same planes are identified (e.g., (422) in Figure 1b,c), a clear trend of the lattice expansion is observed for the bulkier B(III) site (Gd^3+^ vs. In^3+^) in the DHP NC lattice, which is also confirmed by X-ray diffraction (XRD) (Appendix A). XRD also confirms the phase purity of the synthesized DHP NCs. XRD patterns for different NC samples synthesized in this work are in excellent agreement with those for the corresponding bulk references, showing no presence of secondary phases even upon incorporation of Ag+ and Na+ ions.

Selected area electron diffraction (SAED) patterns in Figure 1d–f are consistent with XRD and consist of all even and all odd diffraction spots ((200), (220), (311), (400), (422), (440), and (622)), characteristic of a face-centered cubic structure, confirming the successful formation of the DHP NCs, as illustrated in Figure 1g.

Figure 2 shows the optical properties of typical as-synthesized DHP NCs. Panels a, b, and c in Figure 2 show the absorption (black traces), PL (red traces), and the corresponding photoluminescence excitation (PLE) spectra (dashed blue traces) of CNYC, CNIC, and CNGC NCs, respectively, with a nominal Sb^3+^ doping concentration of 5%. The corresponding absorption spectra exhibit a feature at 315 nm, characteristic of Sb^3+^-doped double-halide perovskite NCs.

This feature fully overlaps with the PLE maxima for the self-trapped exciton (STE) emission, and the lowest lying Sb^3+^ vibronic triplet transition (i.e., ^1^S_0_ → ^3^P_l_) [27], typically observed for metals with the ns^2^ valence configuration (i.e., 5s^2^ for Sb^3+^) in an octahedral crystal field [28]. Although electric dipole selection rules make the remaining Sb^3+^ triplet transitions formally forbidden (i.e., ΔJ = 0 (0 → 0) or ±2), the distinct PLE splitting for the ^3^P_1_ excited state (Figure 2a–c) indicates the presence of its strong phonon coupling, allowing the effective STE and Ln^3+^ excitation channels therein. The origin of the observed luminescence, which varies in spectral position from 440 nm to 480 nm, is attributed to an intrinsic lattice-based self-trapped exciton emission band [29], which is enhanced through coupling between Sb^3+^ triplet excited state and the intrinsic yttrium [30], gadolinium [31], and indium [32] DHP lattice excitations due to their energetic proximities. This is supported by microcrystal or bulk studies on various DHP lattices without Sb^3+^, which show low-intensity luminescence through excitation energies that overlap with those of Sb^3+^. Additionally, optical studies conducted on microcrystals show absorbance features extending into the 400 nm range. In our nanocrystal samples, low-intensity tailing is generally observed in this spectral region; however, some samples exhibit more distinct low-energy absorbance features (see Figure 2a,c), most likely due to differences in scattering during data acquisition.

In addition to steady-state STE emission, delayed emission from Ln^3+^ dopant ions can also be efficiently achieved within these lattices. Panels d, e, and f in Figure 2, corresponding to CNYC, CNIC, and CNGC NCs, respectively, show the delayed luminescence (0.2 ms delay) for Sb^3+^/Dy^3+^/Eu^3+^ co-doped NCs with a nominal doping concentration of 5%/12.5%/12.5% (pink traces). The delayed PLE spectrum of Dy^3+^ exhibits a maximum at ca. 315 nm, with approximately 4 nm shifts across the different DHP NC lattices studied. This feature is intrinsic to the host matrices and is enhanced due to the energetic overlap with the intra-ionic Sb^3+^ transition (^1^S_0_-^3^P_1_), which is formally parity-allowed but spin-forbidden. The comparison of the steady-state and delayed PLE spectra indicates an overlap between the intrinsic STE excitation, sensitized by Sb^3+^, and the intra-ionic f-f emission of Dy^3+^, consisting of electric-dipole (ED) transitions at 484 nm (^4^F_9/2_ → ^6^H_15/2_), 576 nm (^4^F_9/2_ → ^6^H_13/2_), and 755 nm (^4^F_9/2_ → ^6^H_9/2_) and magnetic-dipole (MD) transition at 677 nm (^4^F_9/2_ → ^6^H_11/2_). These results demonstrate an energy transfer between Sb^3+^ and Dy^3+^ excited states, as the most effective pathway for dysprosium sensitization. Meanwhile, Eu^3+^ emission transitions ^5^D_0_ → ^7^F_0_, ^7^F_1_, ^7^F_2_, ^7^F_3_, and ^7^F_4_ (observed at 540, 594, 617, 654, and 703 nm, respectively) show a different excitation mechanism, which does not involve ns^2^ ions (see the discussion related to Figure 3 below). The Eu^3+^ PLE exhibits similar behavior for all DHP host lattices, with the main excitation line located at approximately 297 nm and a broad shoulder at longer wavelengths above 350 nm. This secondary feature may also be ascribed to a Eu^3+^ ligand-to-metal charge transfer (LMCT) transition, as previously observed in Cs_2_NaEuCl_6_ NCs [8].

Additional experiments were conducted to understand further the energy-transfer mechanism involving Sb^3+^-Ln^3+^ ion pairs, probing the excitation and emission pathways of Sb^3+^-Eu^3+^ in varying DHP host matrices. Nanocrystals doped with Sb^3+^/Tb^3+^/Eu^3+^, and Bi^3+^/Ln^3+^/Eu^3+^ (Appendix A) further demonstrate the presence of two distinct energy transfer pathways.

Figure 3 further illustrates this excitation-dependent tuning of the NC emission. Panels a to c show the absorption, PL, and PLE spectra of CNYC, CNIC, and CNGC NCs co-doped with Sb^3+^/Dy^3+^/Eu^3+^ at a nominal doping concentration of 5%/12.5%/12.5%. The PL excitation spectra of Dy^3+^ are nearly identical to those of STEs in the same nanocrystals doped with Sb only (see Figure 1 above). The PLE spectra of Eu^3+^, on the other hand, show a distinct excitation band at ca. 297 nm for all three NC host lattices. Panels d-f show absorption, PL, and PLE spectra of the same host lattices and for the same nominal doping ratios as those in a-c, except that Dy^3+^ is replaced with Tb^3+^. These samples show nearly identical excitation behavior for Tb^3+^ intra-atomic f-f emission, namely ED components at 491 nm (^5^D_4_ → ^7^F_6_), 548 nm (^5^D_4_ → ^7^F_5_), and 622 nm (^5^D_4_ → ^7^F_3_), and MD at 584 nm (^5^D_4_ → ^7^F_4_) as those discussed in panels a-c. The possibility of indirect Ln^3+^ excitation in DHP NCs involving other ns^2^-sensitizing ions (e.g., Bi^3+^), which is particularly effective in the case of CNIC lattice [33], is evident through effective emission of dysprosium (Appendix A), terbium (Appendix A), and erbium (Appendix A) upon excitation of Bi^3+^ s-p transitions. Meanwhile, the excitation pathways of Eu^3+^, as a co-dopant in CNIC NCs appear generally independent of the ns^2^ sensitizer (Appendix A).

Figure 3 further illustrates this excitation-dependent tuning of the NC emission. Panels a to c show the absorption, PL, and PLE spectra of CNYC, CNIC, and CNGC NCs co-doped with Sb^3+^/Dy^3+^/Eu^3+^ at a nominal doping concentration of 5%/12.5%/12.5%. The PL excitation spectra of Dy^3+^ are nearly identical to those of STEs in the same nanocrystals doped with Sb only (see Figure 1 above). The PLE spectra of Eu^3+^, on the other hand, show a distinct excitation band at ca. 297 nm for all three NC host lattices. Panels d-f show absorption, PL, and PLE spectra of the same host lattices and for the same nominal doping ratios as those in a-c, except that Dy^3+^ is replaced with Tb^3+^. These samples show nearly identical excitation behavior for Tb^3+^ intra-atomic f-f emission, namely ED components at 491 nm (^5^D_4_ → ^7^F_6_), 548 nm (^5^D_4_ → ^7^F_5_), and 622 nm (^5^D_4_ → ^7^F_3_), and MD at 584 nm (^5^D_4_ → ^7^F_4_) as those discussed in panels a-c. The possibility of indirect Ln^3+^ excitation in DHP NCs involving other ns^2^-sensitizing ions (e.g., Bi^3+^), which is particularly effective in the case of CNIC lattice [33], is evident through effective emission of dysprosium (Appendix A), terbium (Appendix A), and erbium (Appendix A) upon excitation of Bi^3+^ s-p transitions. Meanwhile, the excitation pathways of Eu^3+^, as a co-dopant in CNIC NCs appear generally independent of the ns^2^ sensitizer (Appendix A).

Although selective excitation of lanthanide dopants is observed for both Dy/Eu and Tb/Eu co-coped DHP NCs, Figure 3 also highlights an essential difference between the PL properties of the two co-doped NC systems. Most notably, in CNIC NC lattice (Figure 3b,e), terbium exhibits much stronger emission than dysprosium, and therefore, dominates over that of europium. Consequently, selective tuning of the spectral density at the parity Tb/Eu ratio is highly limited and requires a lower amount of Tb^3+^ as a co-dopant (see Figure 4). Additionally, an alternative excitation pathway for terbium activation is observed below 250 nm (Figure 3d–f). This excitation channel has been reported for ns^2^-free indium-based DHP NCs [34] and attributed to intrinsic terbium LMCT activation. As we decreased the antimony feed ratio, the inherent activation of terbium became dominant (Appendix A). Moreover, the excitation pathway via LMCT transfer is also available for dysprosium, although to a much smaller extent (Appendix A). Thus, we draw a conclusion that the luminescence of Eu^3+^ occurs almost exclusively via direct excitation involving a LMCT transition [8] (see also Appendix A). In contrast, the luminescence of Tb^3+^, Dy^3+^, or Er^3+^ may also occur through energy transfer from Sb^3+^ (as another co-dopant in DHP lattice) to the lanthanide ions. Finally, we note that the samarium, despite being a relatively bright lanthanide center in DHP lattices [8,35] and showing dual/multiple excitation pathways (Appendix A), is not suitable for selective tuning of PL due to the significant overlap of the main emission lines with those of europium (see Appendix A). Compositional tuning for the emission of Ln^3+^-co-doped CNIC NC system is presented in Figure 4. For the Sb^3+^/Tb^3+^/Eu^3+^-doped CNIC system having a nominal composition of 5%/2.5%/22.5% (Figure 4a), it is possible to realize the strategy described above: Eu^3+^ is activated via intrinsic (LMCT), while Tb^3+^ is activated via the Sb^3+^ (host lattice) excitation pathway. In this work, we did not investigate in depth the compositional tuning of lanthanide-doped DHP NC emission, as reported in the literature for a variety of perovskite and perovskite-like systems (e.g., the Eu/Tb pair [18,36,37]). Our goal was to engineer and control distinct dopant excitation pathways, which was achieved via activation by ns^2^ ions, specifically bismuth and antimony (Figure 4b). The strategy mentioned above is highly effective for white-light rendering in indium-based DHP, as it enables titration of the contributions from “blue” Sb^3+^ and “orange” Bi^3+^ STE emissions [38]. Here, ns^2^ ion excitation introduces an additional channel for Ln^3+^ emission. To quantify the STE (Sb^3+^) to Ln^3+^ energy transfer in DHP NCs, we compared the PL Intensities of Dy^3+^ dopants in CNGC NCs with and without Sb^3+^ as a co-dopant under identical conditions (Appendix A). The Dy^3+^ f-f emission intensity increases by ca. 50-fold upon incorporation of Sb^3+^, demonstrating the critical role that ns^2^ ions play in the energy transfer process. Still, the Eu^3+^ activation pathway remains unaffected by co-doping NCs with ns^2^ ions, offering more opportunities for the selective tuning of the spectral density. The mechanism of selective excitation of Ln^3+^ is presented as a Jablonski diagram in Figure 4c. Selective excitation of lanthanides enabled by different excitation pathways in DHP NCs allows for the precise tuning of the spectral density for specific NC compositions and excitation energies. These properties are particularly promising for anti-counterfeiting applications, where a specific NC composition set by a given authority translates into a unique PL response that the user can easily confirm.

Another level of anticounterfeiting is the time dependence of the signature emissions. Since the PL lifetimes of STE and lanthanide centers in DHP NCs differ significantly (i.e., nanosecond vs. millisecond timescale [25]), time-resolved PL has been proposed as an anti-counterfeiting approach in addition to the excitation-dependent PL of STE and lanthanides [14,15,21]. However, this concept fails for a pair of different lanthanides investigated here due to their very similar emission lifetimes, which are typical of most lanthanide dopants [39] (see, for example, Appendix A for the Eu/Dy pair studies here). For the time-gating concept, a combination of a lanthanide with a transition metal element may be more effective. Specifically, the emissive dopant ions in Tb/Mn-co-doped DHP NC lattices exhibit a notable variation in time decays for Tb^3+^ (4.9 ms) and Mn^2+^ (13.5 ms), as shown in Appendix A, making it applicable for time-gated tuning of spectral density (Figure 5). Only a broad asymmetric band, assigned to Mn^2+^ (^4^T_1_ → ^6^A_1_) ligand-field transition, and sharp Tb^3+^ f-f transitions, are observed in the afterglow PL spectra for different delays. Both dopants in this case are excited via the ns^2^ ion pathway (Appendix A). Figure 5 illustrates how time-gating can be used to realize a specific spectral density for Tb^3+^- and Mn^2+^-co-doped CNIC and CNGC DHP NCs. Due to the lifetime variation, Tb^3+^ emission dominates on a timescale of hundreds of microseconds to a few milliseconds. After a delay of approximately 12 milliseconds, the characteristic orange-red manganese emission becomes the primary phosphorescence feature, providing an additional temporal level of protection against counterfeiting. As both atomic emitters possess similar excitation pathways (Appendix A), authenticity can be verified under the same excitation wavelength probe. It has been suggested that manganese emission therein can be further promoted by energy transfer involving Sb^3+^/Tb^3+^ ions through ^5^D_4_ (Tb^3+^) → ^4^T_1_ (Mn^2+^) [40]; however, the absence of the 240 nm lattice-based excitation line (Appendix A) implies that Mn^2+^ excitation occurs solely through STE (Appendix A). The ability to selectively excite specific ions in the DHP matrices enables precise tuning of spectral density by adjusting the excitation energy, composition, and time-gating readouts.

## 4. Conclusions

In summary, we synthesized a series of colloidal DHP (Cs_2_NaInCl_6_, Cs_2_NaGdCl_6_, and Cs_2_NaYCl_6_) NCs co-doped with different Ln^3+^ ions or a combination of Ln^3+^ and Mn^2+^, and investigated their PL properties in both frequency and time domains. Excitation of the NCs through ns^2^ ions (e.g., Sb^3+^, Bi^3+^) at the host-lattice B(III) sites allows for a strong (ca. 50-fold) enhancement of sensitization of the f-f emission for most Ln^3+^ dopants (e.g., Dy^3+^, Tb^3+^, Er^3+^, Sm^3+^), in addition to the host lattice luminescence. A notable exception is Eu^3+^, which follows an entirely different excitation pathway involving a charge transfer transition. The presence of distinct excitation channels for different Ln3+ dopants in DHP NCs enables selective sensitization of the corresponding f-f emissions, enabling highly sensitive tuning of the PL spectral density via excitation energy (see Appendix A). Expanding on these results, we also demonstrated time-resolved tuning of the afterglow spectral density in the same DHP NCs co-doped with Mn^2+^ and Ln^3+^ ions. Owing to a large difference between the afterglow decay dynamics for Mn^2+^ d-d transition (average PL lifetime of ca. 13.5 ms) and Ln^3+^ f-f transition (average PL lifetime of ca. 5 ms), a broad variation in the PL spectral density as a function of the delay time in the millisecond time regime was achieved for a given excitation energy. While this study predominantly aims to demonstrate the tunability of spectral density in both frequency and time domains for DHP NCs containing multiple dopants, the obtained results are very promising for anti-counterfeiting applications. Specifically, pre-determined PL peak intensity ratios can be assigned by a given authority either in frequency (e.g., NC containing ns^2^ ions together with different Ln^3+^ co-dopants) or time (e.g., NC containing ns^2^ ions together with Ln^3+^ and Mn^2+^ co-dopants) domains. The selected spectral density can then be achieved in DHP NCs having a specific composition, based on multiple sensitization pathways, relative energy transfer efficiencies, and dopant lifetimes. The authenticity of the product can then be verified by simple steady state and/or delayed PL measurements. The results of the present study demonstrate that suitably designed co-doped DHP materials offer a versatile new platform for multilevel anti-counterfeiting and other photonic applications.

## Figures and Tables

**Figure 1 nanomaterials-15-01838-f001:**
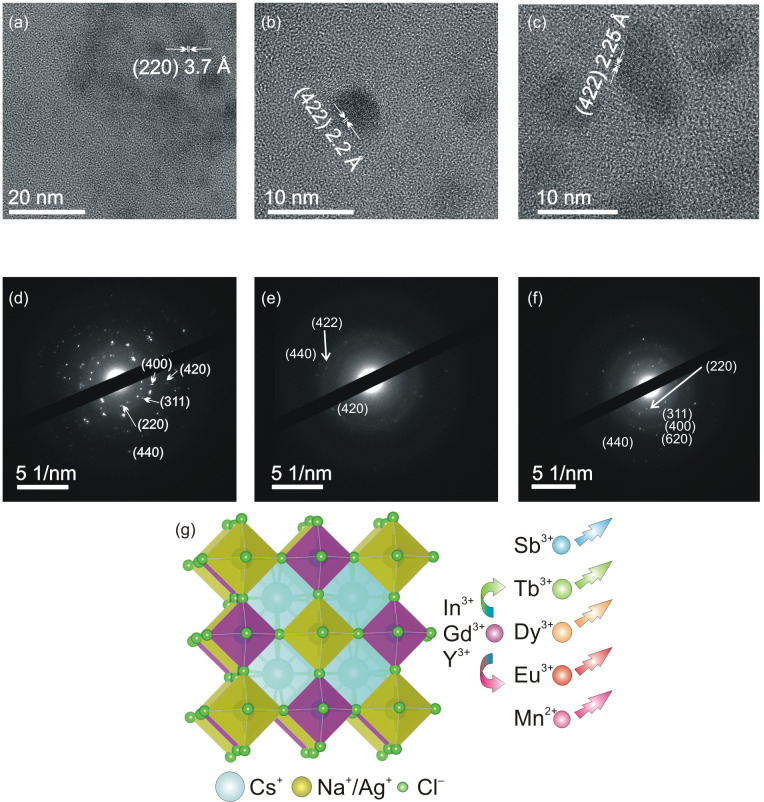
(**a**–**c**) TEM images and (**d**–**f**) SAED patterns of Sb^3+^/Dy^3+^/Eu^3+^ co-doped DHP NCs with nominal doping concentrations of 5%/12.5%/12.5%: CNYC (**a**,**d**), CNIC (**b**,**e**), and CNGC (**c**,**f**). (**g**) Schematic representation of the crystal structure of co-doped DHP NCs.

**Figure 2 nanomaterials-15-01838-f002:**
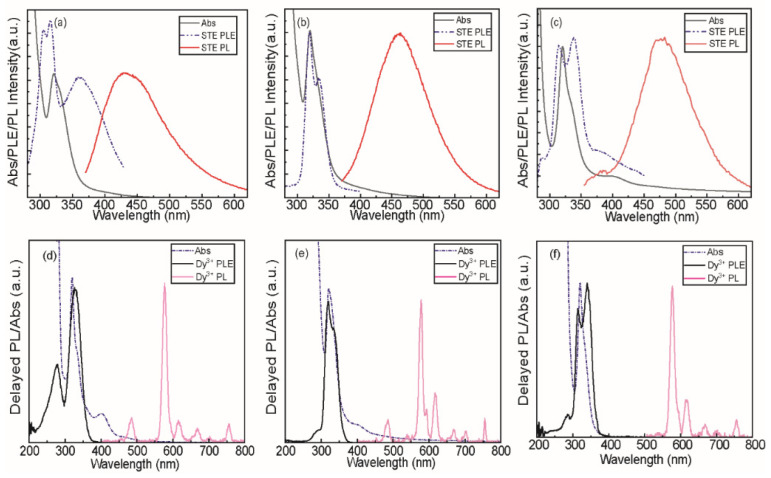
(**a**–**c**) Absorption, steady-state PL (excitation at Sb^3+^ absorbance maximum), and PLE (recorded at STE PL maximum) spectra of DHP NCs with nominal Sb doping concentration of 5% for (**a**) CNYC, (**b**) CNIC, and (**c**) CNGC. (**d**–**f**) Absorption, delayed PL (excitation at Sb^3+^ absorbance maximum), and PLE (recorded for Dy^3+ 4^F_9/2_ → ^6^H_13/2_ emission line at 576 nm) spectra of Sb^3+^/Dy^3+^/Eu^3+^ co-doped DHP NCs with a nominal doping concentration of 5%/12.5%/12.5% for (**d**) CNYC, (**e**) CNIC, and (**f**) CNGC.

**Figure 3 nanomaterials-15-01838-f003:**
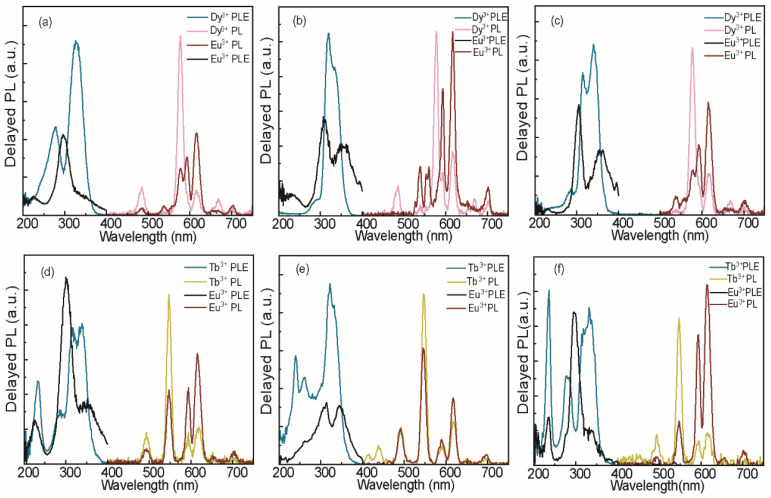
(**a**–**c**) Delayed PL and PLE spectra of Sb^3+^/Dy^3+^/Eu^3+^-co-doped (**a**) CNYC, (**b**) CNIC, and (**c**) CNGC NCs with nominal doping concentrations of 5%/12.5%/12.5%. PL spectra were recorded upon excitation at Sb^3+^ absorbance maximum, with the corresponding PLE spectra recorded for the emission at 576 nm (Dy^3+ 4^F_9/2_ → ^6^H_13/2_ line), and upon excitation at Eu^3+^-Cl^−^ LMCT absorbance maximum at 297 nm with the corresponding PLE spectra recorded for the emission at a 617 nm Eu^3+ 5^D_0_–^7^F_2_ line). (**d**–**f**) Delayed PL and PLE spectra of Sb^3+^/Tb^3+^/Eu^3+^-co-doped (**d**) CNYC, (**e**) CNIC, and (**f**) CNGC NCs with nominal doping concentrations of 5%/12.5%/12.5%. PL spectra were recorded upon excitation at Sb^3+^ absorbance maximum, with the corresponding PLE spectra recorded for the emission at 548 nm (Tb^3+ 5^D_4_ → ^7^F_5_ line), and upon excitation at Eu^3+^-Cl^−^ LMCT absorbance maximum at 297 nm with the corresponding PLE spectra recorded for the emission at a 617 nm Eu^3+ 5^D_0_–^7^F_2_ line).

**Figure 4 nanomaterials-15-01838-f004:**
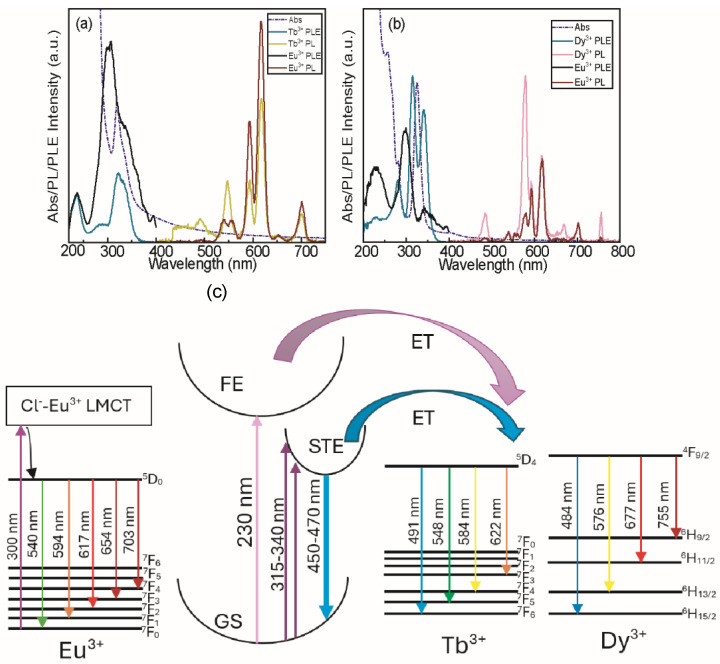
(**a**,**b**) Absorption, delayed PL, and PLE spectra of (**a**) Sb^3+^/Tb^3+^/Eu^3+^-co-doped CNIC NCs with a nominal doping concentration of 5%/2.5%/22.5%, and (**b**) Bi^3+^/Sb^3+^/Dy^3+^/Eu^3+^ co-doped CNIC NCs with nominal doping concentrations of 2.5%/2.5%/12.5%/12.5%. (**c**) Jablonski style diagram showing different excitation pathways in co-doped DHP NCs.

**Figure 5 nanomaterials-15-01838-f005:**
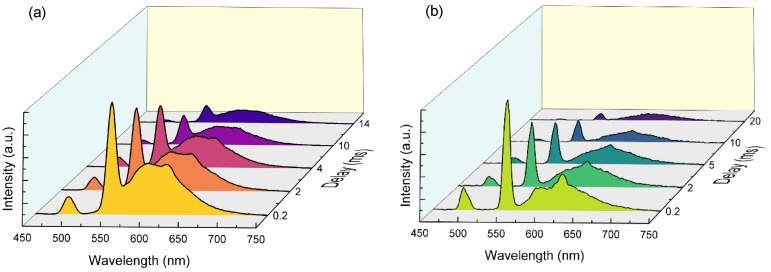
Time-delay dependence of the PL afterglow of Sb^3+^/Tb^3+^/Mn^2+^-co-doped (**a**) CNGC and (**b**) CNIC NCs excited at the Sb^3+^ absorbance maximum. The nominal dopant concentrations of Sb^3+^/Tb^3+^/Mn^2+^ are 5%/16.7%/8.3%.

## Data Availability

The data will be made available upon request.

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
