# Peer review of "Selective Excitation of Lanthanide Co-Dopants in Colloidal Lead-Free Halide Perovskite Nanocrystals as a Multilevel Anti-Counterfeiting Approach"

_nanomaterials, 2025, doi:10.3390/nano15241838_

Round 1
Reviewer 1 Report
Comments and Suggestions for Authors
This manuscript presents a well-designed study on the selective excitation of lanthanide co-dopants in lead-free double halide perovskite nanocrystals for multilevel anti-counterfeiting applications. The work is experimentally rigorous, with systematic synthesis and characterization of co-doped NCs, and demonstrates novel strategies for tuning photoluminescence via excitation pathways and time-gating. The findings are of potential interest to the fields of nanomaterials, photonics, and security materials. However, several issues require clarification or improvement before publication.
- The TEM images reveal distinct lattice fringes, and SAED patterns confirm the face-centered cubic structure of the nanocrystals. However, the absence of size distribution statistics makes it challenging to assess the synthesis homogeneity and reproducibility. It is recommended to supplement with XRD data to verify crystalline phase purity, particularly regarding the structural stability after partial substitution of Ag+ for Na+.
- Figure 2 shows good overlap between the PLE spectra of STE and Ln3+, but the energy transfer efficiency has not been quantified. Can the energy transfer efficiency be provided?
- Recommend conducting relevant data analysis to clarify the Sb3+→Ln3+energy transfer process.
- Only mentions “potential anti-counterfeiting applications” without providing actual demonstrations. Recommend adding simple application scenarios.
- The manuscript requires careful revision on several points.
- Doping concentration descriptions are inconsistent (e.g., 5%/12.5%/12.5% does not specify whether it refers to molar percentage or mass ratio).
- Reference formatting must be standardized.
- Define acronyms (LMCT, STE) upon first use.
- The introduction overly emphasizes prior work on bulk DHPs. Streamline to focus on the nanocrystal-specific advances of this study.
Author Response
This manuscript presents a well-designed study on the selective excitation of lanthanide co-dopants in lead-free double halide perovskite nanocrystals for multilevel anti-counterfeiting applications. The work is experimentally rigorous, with systematic synthesis and characterization of co-doped NCs, and demonstrates novel strategies for tuning photoluminescence via excitation pathways and time-gating. The findings are of potential interest to the fields of nanomaterials, photonics, and security materials. However, several issues require clarification or improvement before publication.
Author reply: We thank the Reviewer for detailed analysis of our manuscript and a favorable assessment of our work and its suitability for publishing in Nanomaterials.
- The TEM images reveal distinct lattice fringes, and SAED patterns confirm the face-centered cubic structure of the nanocrystals. However, the absence of size distribution statistics makes it challenging to assess the homogeneity and reproducibility of the synthesis. It is recommended to supplement with XRD data to verify crystalline phase purity, particularly regarding the structural stability after partial substitution of Ag+ for Na+.
Author reply: In the revised version of the Supporting Information in Figure S2, we included representative XRD patterns for selected DHP NCs studied in this work (i.e., Sb3+/Dy3+/Eu3+-co-doped CNGC and CNIC NCs). As exemplified in new Figure S2, all XRD patterns that we collected are in excellent agreement with those for the corresponding bulk references, showing no presence of secondary phases even upon incorporation of Ag+ and Na+ ions. In the revised version of the manuscript, on page 4, we added a comment regarding XRD data.
“Phase purity of synthesized DHP NCs was confirmed by X-ray diffraction (XRD) measurements. XRD patterns for different NC samples synthesized in this work are in excellent agreement with those for the corresponding bulk references, showing no presence of secondary phases even upon incorporation of Ag+ and Na+ ions.”
- Figure 2 shows good overlap between the PLE spectra of STE and Ln3+, but the energy transfer efficiency has not been quantified. Can the energy transfer efficiency be provided?
Author reply: It is pretty challenging to accurately determine energy transfer efficiencies in these systems due to a substantial overlap between the NC host lattice and ns2 ion peaks in the excitation spectra, possible interdependence of STE and Ln3+ emissions, as well as the lack of a suitable reference for determining relative quantum yield. However, we performed additional experiments (see explanation in response to point 3) to demonstrate an increase in the energy-transfer efficiency to Ln3+ ions in DHP NCs upon incorporation of Sb3+. These data are included in new Fig. S12, and discussed in the main text on page 7.
- Recommend conducting relevant data analysis to clarify the Sb3+→Ln3+energy transfer process.
Author reply: As discussed in the previous point (point 2), we performed additional experiments to address the energy transfer from Sb3+ and Ln3+. Specifically, we synthesized the same NCs (CNGC) doped with only 12.5% Dy3+ and co-doped with 5% Sb3+ and 12.5% Dy3+. The only difference between these two samples is the presence of a small amount of Sb3+, which allowed us to quantify an increase in the energy transfer efficiency due to ns2 ions. We ensured that the concentrations of the colloidal suspensions of these two samples were the same by normalizing the absorbance at 220 nm, corresponding to the host lattice band-to-band transition. Approximately a 50-fold increase in the Dy3+ emission intensity was observed for co-doped NCs, unambiguously demonstrating an order-of-magnitude increase in the energy transfer efficiency due to the presence of ns2 ions. In the revised version of the manuscript, we included a discussion related to this new result on page 7:
“To quantify the STE to Ln3+ energy transfer in DHP NCs, we compared the PL intensities of Dy3+ dopants in CNGC NCs with and without Sb3+ as a co-dopant under identical conditions (Fig. S13). The Dy3+ f-f emission intensity increases by ca. 50 times upon incorporation of Sb3+, demonstrating a critical role that ns2 ions play in the energy transfer process.”
- Only mentions “potential anti-counterfeiting applications” without providing actual demonstrations. Recommend adding simple application scenarios.
Author reply: While this study primarily aims to demonstrate the tunability of the spectral density of DHP NCs containing multiple dopants and the role of selective excitation of different Ln3+ centers in that process, we provided additional clarification of how this phenomenon can be used for anti-counterfeiting applications in the revised manuscript. Specifically, predetermined PL peak intensity ratios can be assigned by a given authority in either the frequency (for NC containing different Ln3+ centers) or time (for NC containing Ln3+ and Mn2+ centers) domain. The specified spectral density can be achieved in DHP NCs with various compositions, depending on multiple sensitization pathways and relative energy-transfer efficiencies. The authenticity of the product can then be verified by simple steady state and/or delayed PL measurements. We added this simple application scenario on pages 11-12 of the revised version of the manuscript:
“While this study predominantly aims to demonstrate the tunability of spectral density in frequency and time domains for DHP NCs containing multiple dopants, the obtained results are very promising for anti-counterfeiting applications. Specifically, pre-determined PL peak intensity ratios can be assigned by a given authority either in frequency (e.g., NC containing ns2 ions together with different Ln3+ co-dopants) or time (e.g., NC containing ns2 ions together with Ln3+ and Mn2+ co-dopants) domains. The selected spectral density can then be achieved in DHP NCs having specific composition, based on multiple sensitization pathways, relative energy transfer efficiencies, and dopant lifetimes. The authenticity of the product can then be verified by simple steady state and/or delayed PL measurements.”
- The manuscript requires careful revision on several points.
- Doping concentration descriptions are inconsistent (e.g., 5%/12.5%/12.5% does not specify whether it refers to molar percentage or mass ratio).
- Reference formatting must be standardized.
- Define acronyms (LMCT, STE) upon first use.
- The introduction overly emphasizes prior work on bulk DHPs. Streamline to focus on the nanocrystal-specific advances of this study.
Author reply: In the revised version, we addressed these points as highlighted in the manuscript. Related to the final point, we note that we cited prior anticounterfeiting-related work on bulk DHPs because there were no relevant results for colloidal NCs. That is, in fact, one of the novel aspects of this work, along with opportunities in device design and fabrication. All the other our introductory references cite only NC-based research.
Reviewer 2 Report
Comments and Suggestions for Authors
Balitskii et al. designed and synthesized lead-free halide perovskite nanocrystals co-doped with a pair of different lanthanides as emission centers and ns² ions as sensitizers. The materials achieve tunable delayed photoluminescence spectral density by modulating excitation wavelength, dopant ratio, and time delay, leveraging distinct decay dynamics of Mn²⁺ d-d and Ln³⁺ f-f transitions. This rational co-doping strategy provides a new platform for pre-designed multilevel luminescent signatures, advancing anti-counterfeiting and photonic applications. The work is well planned and results definitely deserve publication in Nanomaterials. However, there are still some issues that need to be corrected.
- It is suggested that key photophysical parameters should be added in the Abstract and Conclusion sections.
- From the Figure 1 TEM images, the sizes of the nanocrystals seem are uneven and shapes are irregular. It is recommended to provide TEM images with more uniform sizes and more regular shapes. It is also suggested to add a particle size distribution graph.
- It is suggested that the double-exponential fitting results should be supplemented in the main text or the figure notes, and the practical significance of the difference in lifetime for anti-counterfeiting applications should be discussed.
- Please compare the key parameters of these materials with other reported material systems, and emphasize the advantages of these materials.
Author Response
Comments:
Balitskii et al. designed and synthesized lead-free halide perovskite nanocrystals co-doped with a pair of different lanthanides as emission centers and ns² ions as sensitizers. The materials achieve tunable delayed photoluminescence spectral density by modulating excitation wavelength, dopant ratio, and time delay, leveraging distinct decay dynamics of Mn²⁺ d-d and Ln³⁺ f-f transitions. This rational co-doping strategy provides a new platform for pre-designed multilevel luminescent signatures, advancing anti-counterfeiting and photonic applications. The work is well planned and results definitely deserve publication in Nanomaterials. However, there are still some issues that need to be corrected.
Author reply: We thank the Reviewer for the positive assessment of the quality and significance of our manuscript.
- It is suggested that key photophysical parameters should be added in the Abstract and Conclusion sections.
Author reply: The key photophysical parameters that allow for tuning of the spectral density are the efficiency of Ln3+ sensitization by the host lattice in the presence of an ns2 ion and a significant difference in lifetime of Ln3+ and Mn2+ in DHP NCs. While the Abstract is written to be concise and focused, we have included these parameters as quantifiable metrics in the Conclusions section on page 9 of the revised manuscript.
- From the Figure 1 TEM images, the sizes of the nanocrystals seem are uneven and shapes are irregular. It is recommended to provide TEM images with more uniform sizes and more regular shapes. It is also suggested to add a particle size distribution graph.
Author reply: Given the anisotropy of NC sizes (i.e., platelet-like morphology), it is very challenging to obtain TEM images that appear uniform. In that context, providing any statistics would be inaccurate, given the range of NC orientations on the TEM grid. That said, the Reviewer correctly observed that these highly complex NC systems, which contain multiple disparate dopants, are relatively uneven in size and shape. While we acknowledge that NC uniformity is generally an essential parameter from a synthesis perspective, the properties we report are not sensitive to NC size distribution and would not substantially contribute to the manuscript.
- It is suggested that the double-exponential fitting results should be supplemented in the main text or the figure notes, and the practical significance of the difference in lifetime for anti-counterfeiting applications should be discussed.
Author reply: The main reason why we used exponential fitting to estimate the average lifetimes of different dopants in DHP NCs was as a guideline for temporal tuning of the spectral density. These fittings are largely empirical and therefore not relevant to specific photophysical properties. Nevertheless, to comply with the Reviewer's request, we added average lifetimes in the revised version of the main text on page 8 (and Figure S14) to clarify the practical significance of the lifetime difference for anti-counterfeiting applications.
- Please compare the key parameters of these materials with other reported material systems, and emphasize the advantages of these materials.
Author reply: We appreciate the reviewer's comment. In fact, we have studied the most recent review article summarizing Ln emission in solely DHP latices (Journal of Luminescence 277 (2025) 120990). In their summarizing table, authors are generalizing the phenomena and trying to provide proper classification. Still, they could not identify any new parameters, except the PL type (up- or down-conversion), the emitter/ns2 booster, and PLQY. We addressed all those in our paper, except the PLQY measurements. The PLQYs therein combine contributions from STE and lanthanide f-f emission and do not solely reflect lanthanide efficiency. The emphases are provided in the conclusion.